# Microstructure of Dolostones of Different Geological Ages and Dedolomitization Reaction

**DOI:** 10.3390/ma15124109

**Published:** 2022-06-09

**Authors:** Zhiyuan Fan, Zhongyang Mao, Xiang Liu, Lei Yi, Tao Zhang, Xiaojun Huang, Min Deng

**Affiliations:** 1College of Materials Science and Engineering, Nanjing Tech University, Nanjing 211800, China; 201961203172@njtech.edu.cn (Z.F.); mzy@njtech.edu.cn (Z.M.); 201961203150@njtech.edu.cn (X.L.); 201961203130@njtech.edu.cn (L.Y.); 201961203162@njtech.edu.cn (T.Z.); 5967@njtech.edu.cn (X.H.); 2State Key Laboratory of Materials-Oriented Chemical Engineering, Nanjing 211800, China

**Keywords:** geological ages, dolostone, ordering degree, dedolomitization reaction

## Abstract

Dolostone is widely distributed and commonly used as concrete aggregates. A large number of studies have shown that there are significant differences in the expansibility of different dolostones, and the key factors determining the expansibility of alkali carbonate rocks have not been clarified. In this paper, rocks were selected from five different geological ages: Jixianian, Cambrian, Ordovician, Devonian, and Triassic ages. The ordering degree and the content of MgCO_3_ of dolomites in rocks of different geological ages were determined by X-ray diffraction (XRD). The degree of dedolomitization reaction in rocks cured in 80 °C, 1 mol/L NaOH solution was determined by quantitative X-ray diffraction (QXRD). The morphology of dolomites in rocks was determined by a polarizing microscope. The products of the dedolomitization reaction were determined by field emission electron microscopy (FESEM-EDS). According to the test results, the following conclusions are drawn. There is a good positive correlation between ordering degree and the molar fraction of MgCO_3_ of dolomites. When the MgCO_3_ mole fraction of dolomites varies from 47.17% to 49.60%, the higher the MgCO_3_ mole fraction, the greater the ordering degree of dolomite. By analyzing the degree of the dedolomitization reaction of different dolostone powders cured at 80 °C in 1 mol/L NaOH solution, it is found that the older the geological age of dolostone, the slower the dedolomitization reaction rate and the lower the degree of dedolomitization reaction. The lower the ordering degree of dolomite crystal in the same geological age, the faster the rate of dedolomitization reaction and the higher the degree of dedolomitization reaction.

## 1. Introduction

Carbonate rocks are widely distributed throughout China, accounting for about one-fifth of the total surface sedimentary rock distribution [1]. It is often used as aggregate in engineering construction, but due to its alkali dolomite reaction (ADR) activity, it has caused serious damage to engineering construction all over the world, resulting in serious hidden dangers to safety and huge repair costs [2,3]. Therefore, it is of great significance to further study the structure of carbonate rocks. ADR is also known as a dedolomitization reaction. Dedolomitization refers to the calcification of dolomite, a common process by which dolomite is converted to calcite by alkali, which may cause rocks to swell [4]. There are two main types of carbonate rocks: dolostone and limestone [5]. The main mineral of dolostone is dolomite, in addition to a small amount of calcite, quartz, feldspar, and clay minerals. The main mineral of limestone is calcite. The structural characteristics of dolostone mainly include mineral content, grain size, and crystallinity. The molar ratio of Ca^2+^ and Mg^2+^ in ideal dolomite crystals is the same, but Mg^2+^ is often replaced by Ca, Fe, and Mn plasma in nature, which does not conform to the structure of ideal dolomite crystals [6]. The ordering degree is a common method to characterize the crystallinity of dolomite crystals. So far, the intensity ratio of I (015)/I (110) obtained by XRD analysis is the most commonly used way to express the ordering degree of dolomite [7,8,9,10]. Previous studies on the crystallographic characteristics of dolomite have been carried out in detail, but they tend to be pure crystallographic studies, and rarely relate the crystallographic characteristics of different dolomites with their dedolomitization reaction [5,9,11]. Gao [12] analyzed a large amount of data on the ordering degree of dolomite and concluded that the ordering degree of penecontemporaneous dolomite is the worst, and that of deep burial dolomite is the best. Zhong [13] analyzed the Triassic dolomite through XRD, analyzed various factors affecting the ordering degree of dolomite and pointed out that the ordering degree of dolomite formed in the burial environment is the best, while the ordering degree of penecontemporaneous dolomite is the worst. Hadley [14] used XRD to study the relationship between ADR activity of aggregates and crystallinity of dolomite crystals, and the results showed that aggregates with and without ADR activity had poor crystallinity of dolomite, so he pointed out that crystallinity would not affect ADR activity of aggregates. In addition, Deng [15] analyzed the ordering degree of dolomite crystals in different rocks and found that the ordering degree will have a certain impact on ADR activity. Gillott [16,17] and Swenson [17,18] studied dolomite structure, and they found that dolomite grain size had a certain influence on ADR reaction rate. Niu [19] studied dolomite with different grain sizes and found that the larger the grain size of dolomite in dolostone, the higher the degree of dedolomitization reaction. Feng [20] analyzed the ordering degree of dolomite at different burial depths, and the results showed that the ordering degree of dolomite crystals increased with burial depth, while the dedolomitization gradually decreased.

In this study, XRD and polarizing microscope are used to determine the microstructure of dolostones of different geological ages, and the influence of factors such as the ordering degree of dolomite crystal on the degree of dedolomitization reaction was discussed, which provided data support for exploring the ACR expansion law of dolostones. Identifying the types of rocks that are likely to cause damage can lead to the more efficient and rational use of resources.

## 2. Materials and Methods

### 2.1. Raw Materials

#### Rocks

By consulting the 1:50,000 scale geological map of China and referring to the regional geological literature, the dolostones of different geological ages are determined. Twelve rock samples from five geological ages were selected for this experiment, as shown in Figure 1. Among them, the rocks WMS-6 and WMS-8 from Tianjin belong to Jixianian age, the rocks BFL-7 and BFL-12 from Linqu county in Shandong Province belong to Cambrian age, the rocks ZC from Taiyuan belong to Ordovician age, the rocks JF, DH-1 and DH-2 from Baoding belong to Ordovician age, the rocks SFP-1, SFP-2 and SFP-3 from Dushan County in Guizhou Province belong to Devonian age and the rocks DJY from Yibin City in Sichuan Province belong to Triassic age.

The chemical analysis method of rock is carried out according to the standard GB/T 176-2017(CIS, 2017). Table 1 shows the chemical composition of rocks. It can be seen from Table 1 that the MgO content in rocks WMS-6, WMS-8, DH-1, DH-2, SFP-1, SFP-2, SFP-3, and JF is relatively similar, about 20%, the MgO content in rocks DJY and BFL-12 is about 15%, and the MgO content in rocks ZC and BFL-7 is only about 5%. The contents of SiO_2_ are below 10.0% except for rock JF. The content of Fe_2_O_3_ and Al_2_O_3_ in rock DJY is slightly more than 1%.

Figure 2 shows the minerals of the studied dolostones determined by XRD (the scanning speed is 10°/min, the scanning angle is 5~80°). From Figure 2, it can be seen that the main minerals in the dolostones WMS-6, WMS-8, BFL-12, DH-1, DH-2, JF, SFP-1, SFP-2, and DJY are dolomite and a small amount of quartz, while the main minerals in the dolomitic limestone ZC and BFL-7 are dolomite, calcite, and quartz.

### 2.2. Methods

#### 2.2.1. Ordering Degree of Dolomite

The rocks are ground to about 60 μm to avoid damage to crystallinity caused by excessive grinding, dried and then XRD (the scanning angle is 25~40°, the scanning speed is 2°/min) was used to measure the ordering degree of dolomites. XRD data of rocks were analyzed through the Jade software. The integral intensity value I (015) and I (110) of the diffraction peak of the crystal plane of dolomite (015) and (110), respectively, were read [21,22], and ordering degree of dolomites was calculated according to Formula (1): (1)$$\delta =\frac{I\left(015\right)}{I\left(110\right)}$$

The d (104) value of the diffraction peak of dolomite (104) crystal plane on the measurement spectrum was read, and the mole fraction of MgCO_3_ and CaCO_3,_ respectively, was calculated by using Formulas (2) and (3) [23,24]:
Mole fraction of MgCO_3_ (%) = −333.33 d (104) + 1011.99 (2)
Mole fraction of CaCO_3_ (%) = 100% − Mole fraction of MgCO_3_ (%)(3)


#### 2.2.2. Thin Section Petrography

The dolomite grain sizes in rocks of different geological ages were examined by thin slices of different rocks for optical microscope observation. About a hundred pictures were taken of each rock. A polarizing optical microscope (Optiphot-II Pol reflecting light apparatus, 25–400×, Nikon, Tokyo, Japan) with transmitted light was used. The preparation of thin sections was performed according to the section “Thin section specimen preparation” [25].

#### 2.2.3. Dedolomitization Reaction

The dolomite powder was cured in the condition of 80 °C, 1 mol/L NaOH solution, and taken out regularly. After drying, ZnO was used as the internal standard for quantitative analysis by XRD, and then Jade was used to fitting and analyze the XRD pattern to obtain the peak area ratio of ZnO and dolomite, and the content of brucite was calculated by internal standard method. The degree of dedolomitization reaction in rocks is calculated by the ratio of the reduced amount of dolomite to the content of original dolomite, and the scanning speed is 1°/min, the scanning angle is 30~33°. 

#### 2.2.4. Microstructure of Dedolomitization Reaction Products

Firstly, 5~10 mm rock particles were taken out after curing in 80 °C, 1 mol/L NaOH solution for 7 days. Then, rock particles were solidified in resin and polished by vibration polishing machine. After rough grinding, fine grinding, polishing, and drying for 24 h, the samples were observed by FE-SEM. 

## 3. Results

### 3.1. Analysis of Morphology and Distribution Characteristics of Dolomites

Figure 3 shows the distribution of dolomite grains of Jixianian rocks WMS-6 and WMS-8. It can be seen that the dolomite grains are mosaic distribution, which is mainly caused by the existence of a large number of dolomite grains with different grain sizes in the rock. The dolomite crystals are dominated by anhedral crystals. 

Figure 4 shows the distribution of dolomite grains of Cambrian rocks BFL-7 and BFL-12, and the dolomite grains of rock BFL-7 are dispersed in the calcite matrix. The dolomite grains in BFL-12 are mainly distributed in a mosaic state, and a few dolomite grains are dispersed in the calcite matrix. The dolomite grains in BFL-7 and BFL-12 are mainly anhedral with a little subhedral.

Figure 5 shows the distribution of dolomite grains of Ordovician rocks ZC, JF, DH-1, and DH-2. It can be seen from Figure 5a that dolomite grains of rock ZC are dispersed in the calcite matrix. Dolomite grains have a high degree of idiomorphic crystal, and dolomite crystals are mainly euhedral crystals and subhedral crystals. The crystals support each other in a network shape. It can be seen from Figure 5b that the dolomite crystals of rock JF are dominated by anhedral crystals. It can be seen from Figure 5c,d that dolomite crystals of rocks DH-1 and DH-2 are mosaic distribution. The dolomite crystals in rock DH-1 are mainly euhedral crystals and subhedral crystals, while the dolomite crystals in DH-2 are anhedral crystals.

Figure 6 shows the distribution of dolomite grains of Devonian rocks SFP-1, SFP-2, and SFP-3. It can be seen from Figure 6a–c that the dolomite grains in rocks SFP-1, SFP-2, and SFP-3 are mosaic distribution. The dolomite crystals in rocks SFP-1, SFP-2 and SFP-3 are mainly euhedral crystals with a small number of subhedral crystals. 

Figure 7 shows the distribution of dolomite grains of Triassic rock DJY. It can be seen from Figure 7 that the dolomite grains in DJY are mainly mosaic distribution, and a few dolomite grains are dispersed in the calcite matrix. The dolomite crystals are mainly anhedral crystals.

### 3.2. The Ordering Degree of Dolomite

The ordering degree and MgCO_3_ mole fraction of dolomites were analyzed by XRD to characterize the crystallinity of dolomite crystals. Table 2 shows the analysis of rocks of different geological ages by XRD, and the calculation of MgCO_3_ mole fraction and ordering degree of corresponding dolomites. It can be seen from Table 3 that the ordering degree and the content of MgCO_3_ of dolomites are significantly different in rocks of different geological ages.

Figure 8 shows the relationship between the ordering degree of dolomites in rocks of different geological ages and the MgCO_3_ mole fraction of dolomites. As can be seen from Figure 7, the ordering degree of dolomite crystals in Devonian rocks SFP-1, SFP-2, and SFP-3 are the highest, up to 0.83, while the ordering degree of dolomite crystals of Triassic rocks DJY is the lowest, only 0.61. It can also be seen that when the mole fraction of MgCO_3_ varies from 47.17% to 49.60%, there is an approximately linear relationship between the MgCO_3_ mole fraction and the ordering degree of dolomite crystals. The ordering degree of dolomites increases with the increase in the MgCO_3_ mole fraction. When the molar fraction of MgCO_3_ of dolomite crystals is close to 50%, the order degree is higher. When the content of Mg^2+^ and Ca^2+^ in dolomite crystals is close to each other, Mg^2+^ and Ca^2+^ are more likely to be regularly distributed. Dolomite is formed in a low-salinity environment with a low ratio of Mg^2+^ to Ca^2+^, so the crystallization rate of dolomite is relatively slow, so the ordering degree of dolomite is high.

### 3.3. The Degree of Dedolomitization Reaction 

Figure 9 shows the degree of dedolomitization reaction in rocks of different geological ages at different reaction times. It can be seen that there are certain differences in the reaction rates of dolomite in dolostones. The dedolomitization reaction rate of Jixianian rocks WMS-6 and WMS-8 is relatively slow in the early stage. After curing for 4 days, the degree of dedolomitization reaction almost does not change at about 30%. The degree of dedolomitization of the Cambrian rocks BFL-7 and BFL-12 is relatively slow and tends to be stable at about 40%. The dolomite of Ordovician rocks ZC, DH-1, and DH-2 react completely after curing for about 10 days, while the dolomite of Ordovician rocks JF reacted completely at 4 days. The dolomite in Devonian rocks SFP-1, SFP-2, and SFP-3 reacted completely at about 10 days. The dedolomitization reaction rate of Triassic rocks is the highest in the early stage and can be fully reacted at 7 days. The dedolomitization reaction rate of the rocks in the Jixianian and Cambrian ages are the slowest. The dedolomitization reaction rate of Ordovician rocks is slightly faster than that of Devonian rocks and can be fully reacted with the increase in curing time. Therefore, geological age has a certain influence on the dedolomitization reaction in rocks, and the older the geological age of rocks, the lower the degree of dedolomitization reaction and the slower the reaction rate.

### 3.4. Influence of Ordering Degree of Dolomite on Degree of Dedolomitization Reaction

The degree of dedolomitization reaction of different dolostones is different, and the reason for the difference is not clear. By analyzing the ordering degree of dolomites, the reason for the difference in the degree of dedolomitization reaction is explained. 

Table 3 shows the ordering degree of dolomites and the degree of dedolomitization reaction in the dolostones. The ordering degree of dolomite of Jixianian rocks WMS-6 and WMS-8 is 0.74 and 0.78, respectively, and the degree of dedolomitization reaction in WMS-6 is greater than that in WMS-8 at 14 d. The ordering degree of dolomite of Cambrian rocks BFL-7 and BFL-12 is 0.77 and 0.75, respectively. The degree of dedolomitization reaction of dolostone in BFL-7 and BFL-12 is close. The ordering degrees of dolomites of Ordovician rocks ZC, JF, DH-1, and DH-2 are 0.77, 0.72, 0.78, and 0.74, respectively. The dedolomitization rate of dolomites of JF is the fastest, and that of dolomites of DH-1 is the slowest. The ordering degree of dolomites of Devonian rocks SFP-1, SFP-2, and SFP-3 are 0.81, 0.83, and 0.83, respectively. The dedolomitization reaction rate of dolomites in SFP-1 is the fastest, and the dedolomitization reaction rate of dolomites of SFP-2 and SFP-3 are close to each other. The ordering degree of dolomite of Triassic rock DJY is only 0.61, and the dedolomitization reaction rate of rock DJY is the highest. It can be seen that the lower the ordering degree of dolomites in the same geological age, the faster the dedolomitization reaction rate of dolomite, and the higher the degree of dedolomitization reaction.

### 3.5. Microscopic Analysis of Products after Dedolomitization 

Figure 10 shows the SEM-EDS diagram of the products generated after the dedolomitization reaction. It can be seen from Figure 10a,b that a large number of flaky and granular substances are formed around dolomite after the dedolomitization reaction. The results of Figure 10c,d EDS show that flaky substance (point 1) is mainly composed of Mg and O, which can be judged as brucite after dedolomitization reaction, while granular substance (point 2) is mainly composed of Ca, C and O, which is another product calcite after de dolomitization reaction. The shape of calcite is mostly granular and columnar, and the shape of brucite is mainly flake. These reaction products are distributed and accumulated around dolomite crystals, and it can be seen that there are a large number of pores between brucite and calcite.

## 4. Discussion 

The influence of the morphology of dolomite on the dedolomitization reaction was discussed. The dolomites of rocks WMS-6, WMS-8, Bfl-7, Bfl-12, JF, DH-2, and DJY are mainly anhedral crystals. Among them, the degree of dedolomitization reaction of dolostones WMS-6, WMS-8, BFL-7, and BFL-12 is relatively low, and the degree of dedolomitization reaction is only about 40% at 14 d. The degree of dedolomitization reaction of dolostones DH-2 and DJY is close to 100% at 7 days. The degree of dedolomitization reaction of dolostone ZC, SFP-1, SFP-2, and SFP-3 with a good euhedral degree is close to 100% at about 14 d, and there is a great difference in the degree of dedolomitization reaction at the initial stage of dedolomitization reaction, among which, the degree of dedolomitization reaction of rock ZC reaches 59.1% at 4 d. However, the degree of dedolomitization reaction of rock SFP-2 is only 31.6% in 4 d. There is no correlation between the crystal morphology of dolomites and the degree of dedolomitization reaction, so the crystal morphology of dolomite in dolostones is not the key factor to determine the degree of dedolomitization reaction.

## 5. Conclusions

Through the study of rocks’ microstructure and dedolomitization reaction in dolostones of different geological ages, the conclusions are as follows.
The ordering degree of dolomite crystals in Devonian rocks is the highest, which is 0.83. The ordering degree of dolomite crystals in Jixianian, Cambrian, and Ordovician rocks is about 0.75. The ordering degree of dolomite crystals in Triassic rocks is the lowest, which is 0.61. The MgCO_3_ mole fraction of dolomite crystals in the test rocks ranges from 0.4717 to 0.4960. With the increase in the MgCO_3_ mole fraction, the ordering degree of dolomite increases, and the relationship between the MgCO_3_ mole fraction and the ordering degree is approximately linear.After curing in a 1 mol/L NaOH solution at 80 °C, the dedolomitization reaction rate of dolomite in 0.045–0.080 mm Triassic rock powders is the fastest, and the dolomite completely reacts at 7 days. The dedolomitization rate in the Jixianian rock powder sample is the slowest, and the dedolomitization reaction degree reaches about 40% at 7 days, and then hardly changes. The dedolomitization reaction degree of the dolomite in the Cambrian rock powder sample reaches about 50% at 10 d and then tends to be stable. The dedolomitization reaction rates of powder samples in Ordovician and Devonian rocks are similar, and the dolomites in the rocks can fully react after 10 days and 14 days, respectively. Therefore, it can be inferred that the older the geological age of rocks, the slower the dedolomitization reaction rate, and the lower the degree of dedolomitization reaction.The effects of the ordering degree of dolomites on the degree of dedolomitization reaction in different dolostones are analyzed. The lower the ordering degree of dolomite crystals in rocks of the same geological age, the faster the rate of dedolomitization reaction and the higher the degree of dedolomitization reaction.The products of the dedolomitization reaction of dolostones were determined by SEM-EDS, and the calcite and brucite were distributed around the dolomite crystals, and there were many tiny pores between the calcite and brucite.

## Figures and Tables

**Figure 1 materials-15-04109-f001:**
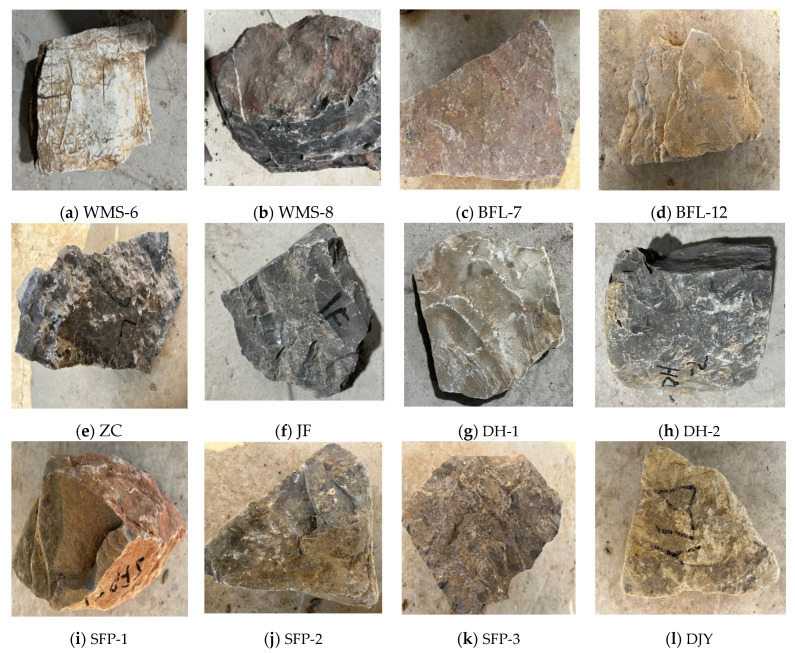
The appearance of dolostones.

**Figure 2 materials-15-04109-f002:**
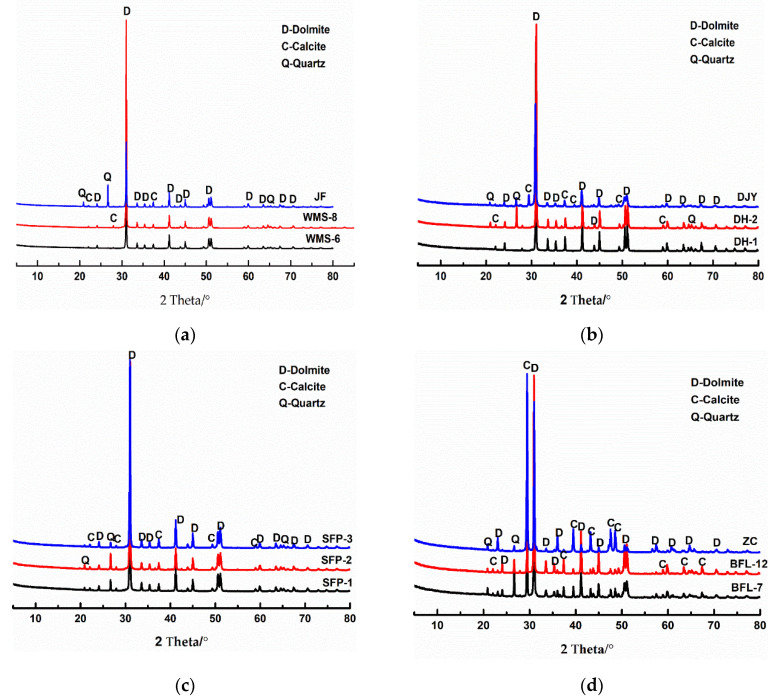
XRD patterns of rocks: (**a**–**c**) dolomite; (**d**) dolomitic limestone.

**Figure 3 materials-15-04109-f003:**
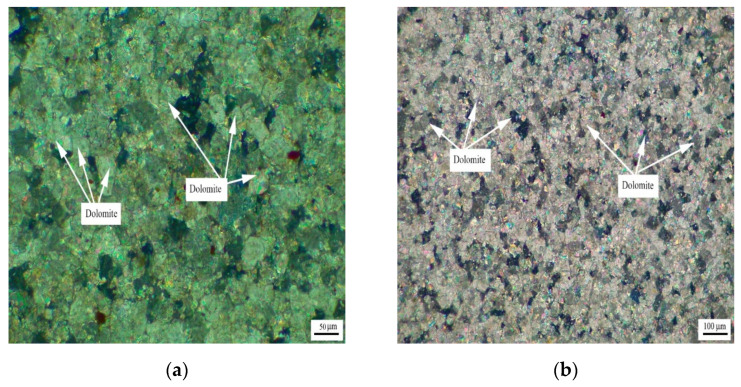
Distribution of dolomites in Jixianian rocks (**a**) WMS-6 and (**b**) WMS-8.

**Figure 4 materials-15-04109-f004:**
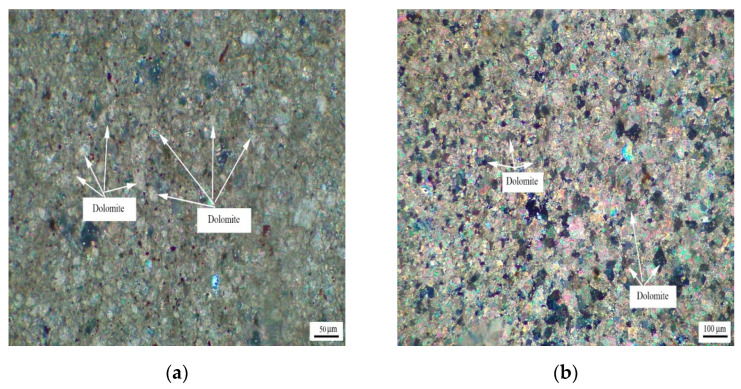
Distribution of dolomites in Cambrian rocks (**a**) BFL-7 and (**b**) BFL-12.

**Figure 5 materials-15-04109-f005:**
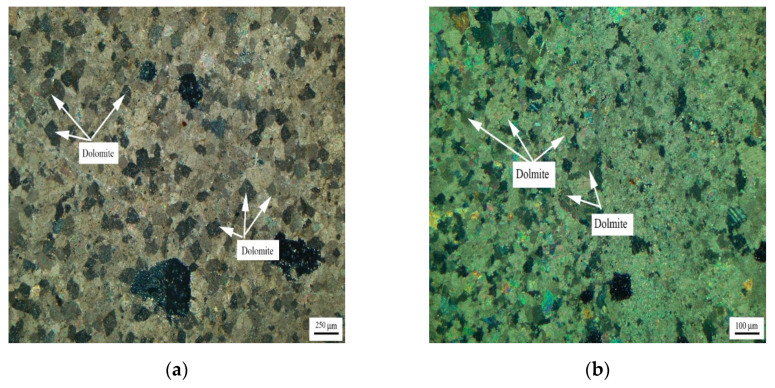
Distribution of dolomites in Ordovician rocks (**a**) ZC, (**b**) JF, (**c**) DH-1, and (**d**) DH-2.

**Figure 6 materials-15-04109-f006:**
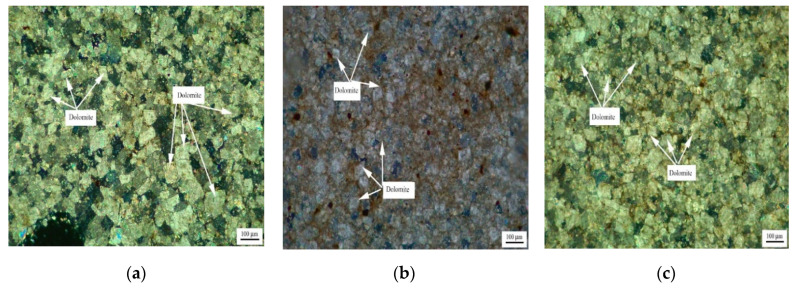
Distribution of dolomites in Devonian rocks (**a**) SFP-1, (**b**) SFP-2, and (**c**) SFP-3.

**Figure 7 materials-15-04109-f007:**
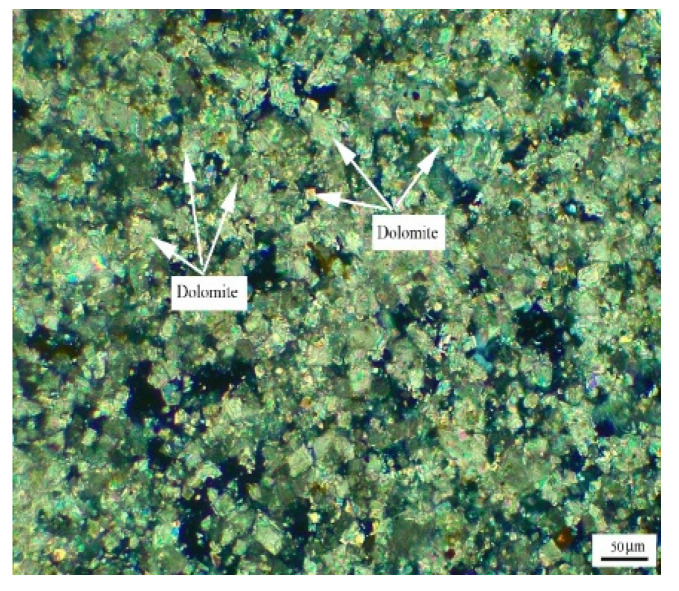
Distribution of dolomites in Triassic rock DJY.

**Figure 8 materials-15-04109-f008:**
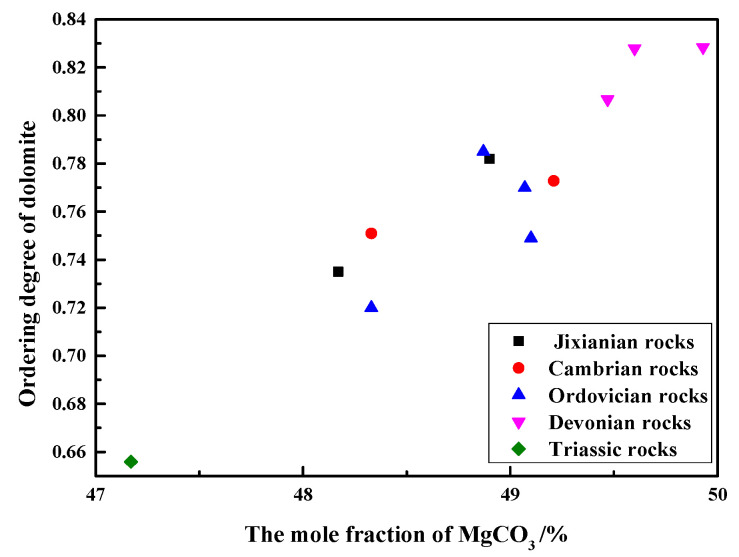
Relationship between ordering degree and the mole fraction of MgCO_3_ of dolomites.

**Figure 9 materials-15-04109-f009:**
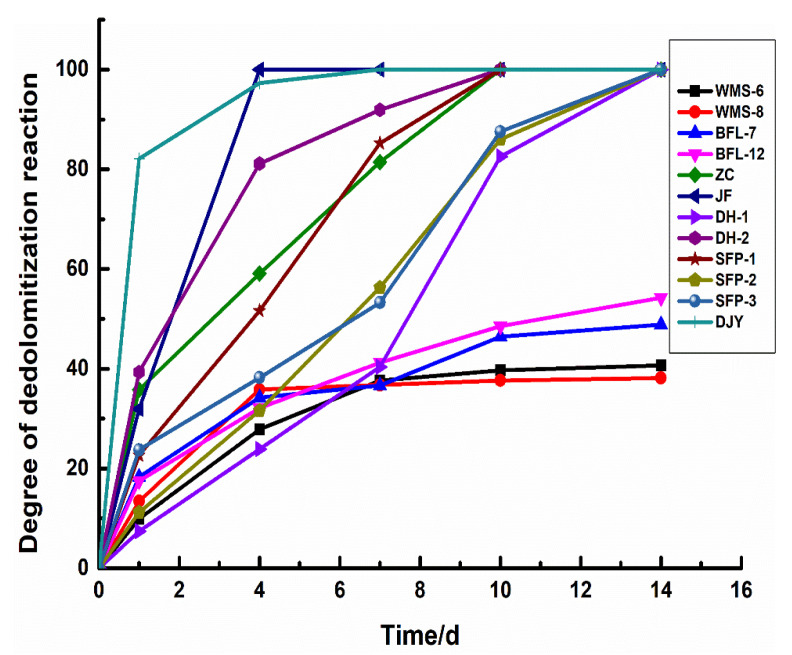
The degree of dedolomitization reaction in rocks.

**Figure 10 materials-15-04109-f010:**
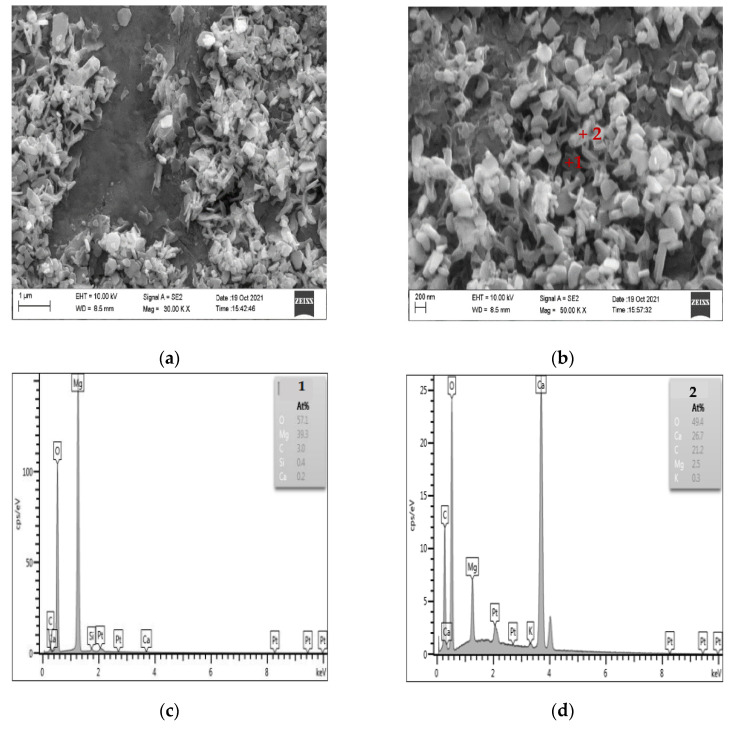
SEM-EDS images of dedolomitization reaction products. (**a**,**b**) The secondary electron image of dedolomitization reaction product (**c**) The elements composition of point 1 (**d**) The elements composition of point 2.

**Table 1 materials-15-04109-t001:** Chemical composition of rocks.

Geological Age	Rocks	Chemical Composition/%								
		SiO_2_	CaO	MgO	Al_2_O_3_	Fe_2_O_3_	K_2_O	Na_2_O	Loss	Total
Jixianian	WMS-6	0.18	29.99	22.61	0.12	0.32	0.03	0.07	46.52	99.84
	WMS-8	6.16	26.56	19.77	0.41	1.22	0.03	0.07	45.48	99.70
Cambrian	BFL-7	2.68	44.04	4.81	0.50	0.22	0.08	0.12	47.11	99.56
	BFL-12	3.38	34.74	15.86	0.84	0.52	0.06	0.13	43.26	98.79
Ordovician	ZC	2.38	47.03	5.13	0.24	0.33	0.13	0.18	43.56	98.58
	JF	10.51	26.50	19.10	0.24	0.33	0.03	0.08	41.2	97.99
	DH-1	0.63	29.13	21.85	0.24	0.33	0.04	0.04	46.11	98.37
	DH-2	4.86	25.34	18.06	0.25	0.32	0.12	0.09	47.81	96.85
Devonian	SFP-1	3.83	28.38	20.53	0.25	0.50	0.09	0.11	44.28	97.97
	SFP-2	1.31	29.23	20.67	0.25	0.50	0.19	1.17	42.81	96.13
	SFP-3	0.18	30.58	20.64	0.13	0.25	0.41	0.15	45.62	97.96
Triassic	DJY	6.78	29.83	16.47	1.04	1.49	0.73	0.23	42.31	98.88

**Table 2 materials-15-04109-t002:** The MgCO_3_ mole fraction and ordering degree of dolomite crystals in rocks.

Rocks	Geological Age	d (104) /Å	Ordering Degree I (015)/I (110)	Mole Fraction/ mol/%	
				CaCO_3_ MgCO_3_	
WMS-6	Jixianian	2.8915 ± 0.0532	0.7352 ± 0.0883	51.83	48.17
WMS-8	Jixianian	2.8893 ± 0.0611	0.7821 ± 0.0774	51.10	48.90
BFL-7	Cambrian	2.8910 ± 0.0722	0.7710 ± 0.1103	51.67	48.33
BFL-12	Cambrian	2.8884 ± 0.0631	0.7528 ± 0.0757	50.79	49.21
ZC	Ordovician	2.8888 ± 0.1050	0.7694 ± 0.0934	50.93	49.07
JF	Ordovician	2.8910 ± 0.0611	0.7182 ± 0.1032	51.77	48.33
DH-1	Ordovician	2.8894	0.7789	51.13	48.87
DH-2	Ordovician	2.8887 ± 0.0933	0.7394 ± 0.0952	50.90	49.10
SFP-1	Devonian	2.8876	0.8067	50.53	49.47
SFP-2	Devonian	2.8862 ± 0.0612	0.8284 ± 0.0544	50.07	49.93
SFP-3	Devonian	2.8872 ± 0.0403	0.8279 ± 0.0448	50.40	49.60
DJY	Triassic	2.8945 ± 0.0972	0.6059 ± 0.1176	52.83	47.17

**Table 3 materials-15-04109-t003:** The ordering degree of dolomite and degree of dedolomitization reaction.

Rocks	The Ordering Degree of Dolomite	The Degree of Dedolomitization Reaction/%			
		1 d	4 d	7 d	14 d
WMS-6	0.74	9.92	27.86	37.66	40.66
WMS-8	0.78	13.45	35.83	36.76	38.16
BFL-7	0.77	18.34	34.25	36.67	48.88
BFL-12	0.75	17.54	32.11	41.24	54.23
ZC	0.77	35.79	59.10	81.44	100.00
JF	0.72	31.94	78.00	100.00	100.00
DH-1	0.78	7.44	23.92	40.39	100.00
DH-2	0.74	39.34	81.15	91.92	100.00
SFP-1	0.81	22.63	51.62	85.24	100.00
SFP-2	0.83	11.21	31.61	56.28	100.00
SFP-3	0.83	23.75	38.20	53.27	100.00
DJY	0.61	82.10	97.30	100.00	100.00

## Data Availability

The data presented in this study are available on request from the corresponding author.

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
