# Peer review of "Microstructure of Dolostones of Different Geological Ages and Dedolomitization Reaction"

_materials, 2022, doi:10.3390/ma15124109_

Round 1

Reviewer 1 Report

The manuscripts needs English revision. In the abstract at the end indicate the novelty of the paper. at the end of the introduction stress the originality of the paper. Picture of the samples would be nice to be included. Figures 1-6 could be merged in one Figure for rapid visualization and comparison. Statistic significance of data in table 2 should be given.

Reviewer 2 Report

Fan et al. Investigated 12 dolostone samples from 5 different geological ages by means of polarizing microscopy, XRD analysis, FE-SEM-EDS analysis, XRF analysis, nitrogen adsorption and “dedolomitization” reaction. The ordering degree of dolomite, cell parameters, “dedolomitization” parameters (degree and rate) and textural information are derived from the results and some mutual relationships established and listed as a conclusion. As a general comment, I would say that’s all. The main flaw of the manuscript, especially the conclusions, lies in the lack of broader framework in which the results should be placed. This leaves a feeling of unfinished work, that the results have not been fully exploited. Just for inspiration : the fact that the older the dolomite, the higher the ordering degree, the closer the cell parameter to ideal dolomite, the slower the “dedolomitization” rate, etc., are all expressions of the same effect of diagenesis during burial in the sedimentary basin, i.e. progressive stabilization of dolomite with time, and the few observed deviations (like for Devonian samples), the effect of initial ordering due to different crystallization conditions (fast or slow, under high or low-salinity, etc.) Finally, I still have a problem with the “dedolomitization” reaction. What’s the point with this method? What is the added value compared to the other methods?

Having reviewed the first version of this manuscript, I feel a bit disappointed by this newer version. While the manuscript has been erratically improved, with some new – and welcome - experiments, a number of my remarks have not been taken into account. To point only a few :

Line 110 : who is Jade? I guess it is not the name of the person who has analyzed your spectra (could be the meaning of your sentence), but a (commercial) software. Again, clarify this.

Page 5, 6 and 7 : the labels in the figure should be “dolomite” and not “Dolmite”! Again, are these cumbersome labels necessary?

Perhaps many of these remarks are details but many details make a load…

See the original annotated manuscript and check for other remarks.

It is also surprising that two methods are inspired from the cement research/industry : bulk chemical analysis and thin-section procedures. By the way, for chemical analysis, citing the standard code without the basic method is a bit short. Please, provide the reader with a minimum of intelligible information, this is scientific communication not technical laboratory report.

Reviewer 3 Report

Dear editor, I have been concluded my review of the manuscript Microstructure of dolostones of different geological ages and dedolomitization reaction from Fan et al., 

First of all, the manuscript addresses a critical topic for material sciences: the aggregate materials in engineering. A topic that is sometimes left out but is vital for most of the civil engineering construction. The authors focus on the relation between dedolomitization process and the age of the rocks. Regarding this, they achieved good results. However, I believe that they could improve the manuscript's impact and quality in several aspects.

First of all, the manuscript is generally well written, and only minor adjustments need to be made. I have done a few of them in the manuscript – pdf attached. 

The main problem is the lack of commas  – Please follow this rule – 

Use commas to separate words and word groups in a simple series of three or more items. 

Another minor problem is the overuse of dedolomitization. – The word is repeated too much in the abstract, for example.  

Regarding the general structure of the manuscript, the Abstract must be improved. Please follow the following model –

A good abstract should present:

1. Your research problem and objectives (that are not present). 

2. Your methods 

3. Your key results or arguments

4. Your conclusion

There is no any reference for the research problem or objectives in the abstract. Also, there is an overuse of the word dedolomitization in the text. 

I could see the research problem addressed in the introduction, so only a minor effort will needed to include it in the abstract. 

Regarding the whole manuscript:

A map for the geological or even geographical location of the samples should be necessary. 

Also, there is some confusion concerning the methods and results –table 1 is a results table. I know that the chemical composition of the rocks are not the main result of the manuscript; however, they are also results. 

I believe that two other issues must be fixed and could improve the significance of the manuscript. 

Regarding the bibliography cited for manuscript construction, I believe that an effort to improve the references should be made. For example, only one manuscript is cited referring to the aggregate relation with damage in structures. 

Materials are not my primary expertise; however, I could find several studies that could reinforce the manuscript conclusions. For sure many important studies have been left out, and I believe that for more robust conclusions, it must be done. 

Another important topic left aside is the geological context of the studied samples. How buried are and were these rocks?; How is the deformation degree of these rocks? It must be addressed to increase de robustness of the study. 

The deposition time is a linear variable and easy to track, but the rock history from their deposition time to nowadays is an entirely different problem. I know that the manuscript topic is time x dedolomitization, but I believe that an effort to at least introduce these variables must be made.  

So, after the revision, I would recommend reconsidering after major revision. As mentioned above, the same minor English and format must be made (abstract, materials, and results) and in some aspects, a more robust discussion must be made. 

Best regards. 

Round 2

Reviewer 3 Report

Dear editor, 

After revising the revised manuscript, I believe that the main points were fixed or explained. Based on this, I recommend accepting the manuscript in its present form. 

Best regards, 

This manuscript is a resubmission of an earlier submission. The following is a list of the peer review reports and author responses from that submission.

Round 1

Reviewer 1 Report

The English form is almost incomprehensible, the subject of the study is vaguely defined. The applied experimental methods are not applied correctly and are not clearly commented in the text.
The results obtained are insignificant and of no interest to the international
scientific community. For detailed comments see the attached pdf file.
My opinion is to reject the manuscript, which is absolutely not suitable
for publication in an international scientific journal such as Materials.

Reviewer 2 Report

In my opinion manuscript materials-1669471 is well written and deserves publication after revison.

The main question addressed is the effect of geological age on microstructure and dedolomitization reaction of dolomite. The results show that there is a good positive correlation between ordering degree of dolomite and the molar fraction of MgCO3 when the molar fraction of MgCO3 is 48 ~ 50% in the same geological age dolomite. Also, it was found that the older the geological age, the less the degree of dedolomitization reaction.

Some suggestion to improve the paper:

  1. At the end of the abstract the novelty of the paper should be emphasized.
  2. English revision and typos corrections are strongly recommended.

For example phrase: “The results show that there is a good positive correlation between ordering degree of dolomite when the molar fraction of MgCO3 in dolomite is 48%~50% and the molar fraction of MgCO3 in dolomite in the same geological age.” Could be rephrased: “The results show that there is a good positive correlation between ordering degree of dolomite and the molar fraction of MgCO3 when the molar fraction of MgCO3 is 48~50% in the same geological age dolomite.”

  • rocks should be Rocks

line 182 and through the text add space after number and unit. um should be mm.

  1. Abbreviation should be given where first appear: see ADR pg 2, line 48, BJH in table 3
  2. Novelty and originality of the paper should be given at the end of the Introduction
  3. In section 2.1.1. give a map of samples provenience and the time scale of the samples age. Indicate the provenience of the rocks, geological collection…
  4. Content from line 79 to line 89 should be moved to Results section. Explain chemical composition calculation of table 1.
  5. At section 2.2 give the XRD analysis details.
  6. At 2.2.1 indicate how was obtained the rock powder.
  7. At 2.2.2. describe the instruments used for analysis.
  8. Text from line 143 to 147 “Fig-…… in the rock.” appears also from line 149 to line 152.
  9. Increase the size of fig. 2
  10. At 3.2. The size porosity should be Rock porosity
  11. At Fig 3. Correct Dolmite, increase picture resolution and text legibility
  12. Combine Fig 5 a and b in one 3D plot with ordering degree of dolomite on vertical axis and dolomite concentration and molar percentage of MgCo3 on horizontal axes for better visualization.
  13. Novelty of the paper should be emphasized in conclusions.
  14. Include references from the last 10 years in the bibliography.

Reviewer 3 Report

Dear Authors

The English of the manuscript is very poor and disappointing! it is most likely literal translation using google translation from the mother tangue of the authors to English. This very obvious in the case of scientific terms. I have attached annotated pdf showing examples particulalrly in the Abstract and Intoduction sections.

Reviewer 4 Report

General comment :

There is a terminological problem with the main material in the paper : “dolomite” is a term referring to the mineral with theoretical formula CaMg(CO3)2. To avoid confusion between the rock-forming mineral and the mineral, a rock containing >50% dolomite must be called “dolostone”. There are also terminological issues when the authors describe the structure of the rocks. They must follow the petrological/petrographical terminology.

The introduction should be reworked, especially by defining dedolomitization (which has different meanings) and explain why it could be relevant for the study.

The “Materials and Methods” section is incomplete (no information on general geology, equipment and settings as standard in scientific publication, no information neither on grain size measurement under the microscope) and mixes results and methods. See annotated manuscript pdf.

Tables and figures must be improved significantly, especially by sorting samples consistently and adding the geological age, which is claimed important for the conclusion.

The lack of geological information, especially the possible origin of dolomite in the different samples has a potential impact on the robustness of the discussion and the conclusion.

Specific comments : see annotated pdf
